# Monkey EEG links neuronal color and motion information across species and scales

**Florian Sandhaeger**[1,2,3,4]*, **Constantin von Nicolai**[1,2,3], **Earl K Miller**[5], **Markus Siegel**[1,2,3]*

[1]Centre for Integrative Neuroscience, University of Tübingen, Tübingen, Germany; [2]Hertie Institute for Clinical Brain Research, University of Tübingen, Tübingen, Germany; [3]MEG Center, University of Tübingen, Tübingen, Germany; [4]IMPRS for Cognitive and Systems Neuroscience, University of Tübingen, Tübingen, Germany; [5]The Picower Institute for Learning and Memory and Department of Brain and Cognitive Sciences, Massachusetts Institute of Technology, Cambridge, United States

**Abstract** It remains challenging to relate EEG and MEG to underlying circuit processes and comparable experiments on both spatial scales are rare. To close this gap between invasive and non-invasive electrophysiology we developed and recorded human-comparable EEG in macaque monkeys during visual stimulation with colored dynamic random dot patterns. Furthermore, we performed simultaneous microelectrode recordings from 6 areas of macaque cortex and human MEG. Motion direction and color information were accessible in all signals. Tuning of the non-invasive signals was similar to V4 and IT, but not to dorsal and frontal areas. Thus, MEG and EEG were dominated by early visual and ventral stream sources. Source level analysis revealed corresponding information and latency gradients across cortex. We show how information-based methods and monkey EEG can identify analogous properties of visual processing in signals spanning spatial scales from single units to MEG – a valuable framework for relating human and animal studies.

DOI: https://doi.org/10.7554/eLife.45645.001

**\*For correspondence:**
florian.sandhaeger@uni-tuebingen.de (FS);
markus.siegel@uni-tuebingen.de (MS)

**Competing interests:** The authors declare that no competing interests exist.

## Introduction

How do results from human magnetoencephalography (MEG) and electroencephalography (EEG) experiments relate to those obtained from animals in invasive electrophysiology? It is generally well understood how potential changes in large populations of neurons can propagate through tissue types and lead to detectable electric potentials and associated magnetic fields outside the head (*Pesaran et al., 2018*). Yet, in typical MEG and EEG experiments, we have little clue which specific cellular and circuit mechanisms contribute to the recorded signals (*Cohen, 2017*).

This can be attributed to several factors. First, the reconstruction of cortical sources from non-invasive signals is generally limited and based on assumptions (*Darvas et al., 2004*). Second, invasive and non-invasive electrophysiology are largely separate research fields. Comparable experiments performed on both levels and in the same species are rare, with few recent exceptions (*Bimbi et al., 2018*; *Godlove et al., 2011*; *Reinhart et al., 2012*; *Shin et al., 2017*; *Snyder et al., 2015*; *Snyder et al., 2018*). Third, studies employing invasive and non-invasive methods in parallel suffer from sparse sampling of recording sites. Massively parallel invasive recordings in multiple brain regions have only recently become viable (*Dotson et al., 2017*; *Jun et al., 2017*; *Siegel et al., 2015*), and EEG recordings in awake behaving animals have so far been limited to relatively few

**eLife digest** Neurons carry information in the form of electrical signals, which we can listen to by applying sensors to the scalp: the resulting recordings are called an EEG. Electrical activity within the brain also generates a weak magnetic field above the scalp, which can be measured using a technique known as MEG. Both EEG and MEG only require a few dozen sensors, placed centimeters away from the brain itself, but they can reveal the precise timing and rough location of changes in neural activity.

However, the brain consists of billions of neurons interconnected to form complex circuits, and EEG or MEG cannot reveal changes in activity of these networks in fine detail. In animals, and in patients undergoing brain surgery, scientists can use hair-thin microelectrodes to directly record the activity of individual neurons. Yet, it is difficult to know how activity measured inside the brain relates to that measured outside.

To find out, Sandhaeger et al. had monkeys and healthy human volunteers perform the same task, where they had to watch a series of colored dots moving across a screen. The brain of the human participants was monitored using MEG; in the monkeys, EEG provided an indirect measure of brain activity, while microelectrodes directly revealed the activity of thousands of individual neurons.

All three recordings contained information about movement and color. Moreover, the monkey EEG bridged the gap between direct and indirect recordings. Sandhaeger et al. identified signals in the monkey EEG that corresponded to the microelectrode recordings. They also spotted signals in the human MEG that matched the monkey EEG. Linking non-invasive measures of brain activity with underlying neural circuits could help to better understand the human brain. This approach may also allow clinicians to interpret EEG and MEG recordings in patients with brain disorders more easily.
DOI: https://doi.org/10.7554/eLife.45645.002

electrodes. This sparsity limits specificity when drawing conclusions from one level to the other. In summary, the mapping between measurement scales is severely underconstrained, both theoretically when trying to infer cortical sources of non-invasively measured activity, and experimentally by the lack of sufficiently comparable data.

Thus, key for linking different scales are comparable large-scale recordings on all levels to provide high specificity and eventually trace the origins of large-scale phenomena back to their underlying cellular mechanisms. Importantly, this includes non-invasive recordings in animals. These allow to bridge the gap between invasive animal electrophysiology and non-invasive human experiments by permitting to disentangle similarities and differences due to species membership from those due to measurement technique. An especially suitable candidate for this is monkey EEG, making use of evolutionary proximity and promising to better connect the rich literature in non-human primate neurophysiology with human studies.

A powerful tool to link data from different measurement scales is the abstraction from measured activity itself to its information content, as enabled by multivariate decoding methods. Representational similarity analysis (RSA) compares the representational structure of signals (*Cichy et al., 2014*; *Kriegeskorte et al., 2008a*). However, as decoding approaches have inherent difficulties to identify the sources of decodable information (*Carlson et al., 2018*; *Liu et al., 2018*), it is necessary to employ thoughtful control analyses or experiments (*Cichy et al., 2015*) to disambiguate different possible mechanisms underlying large-scale information structure. This crucially relies on empirical knowledge about processes on the circuit-scale.

To bridge the gap between invasive and non-invasive electrophysiology, in the present study, we developed and employed fully human-comparable high-density monkey EEG. We presented identical color and motion stimuli to both human participants and macaque monkeys and combined large-scale recordings on multiple scales, including invasive electrophysiology from six areas across the macaque brain, monkey EEG and human MEG with multivariate decoding and representational similarity analysis. We found color and motion direction information not only in invasive signals, but also in EEG and MEG. We show how motion and color tuning in human MEG can be traced back to the properties of individual units. Our results establish a proof of principle for using large-scale

electrophysiology across species and measurement scales to link non-invasive recordings to circuit-level activity.

## Results

To compare information about color and motion direction in invasive and non-invasive signals, we presented rapid streams of dynamic random dot kinematograms (RDKs) with varying color and motion direction to macaque monkeys and humans (*Figure 1A*). We measured single-unit activity, analog multi-unit activity and local field potentials from multiple microelectrodes in six areas of two macaques, and MEG in eleven human volunteers. In order to establish a link between these data that differed both in species model and measurement technique, we developed non-invasive, human-comparable macaque EEG (for details, see methods). We used custom-made 65-channel caps with miniaturized EEG electrodes to measure scalp-EEG in two animals. This data, matching the invasive recordings in terms of species and the human MEG in terms of signal scale, allowed to relate circuit-level activity to large-scale measurements in humans. After preprocessing, we treated all data types identically and submitted them to the same multivariate pattern analysis (MVPA) of visual information. We used multi-class LDA (*Hastie et al., 2009*) and a cross-validation scheme to derive time-resolved confusion matrices. For each combination of two stimulus classes A and B, the

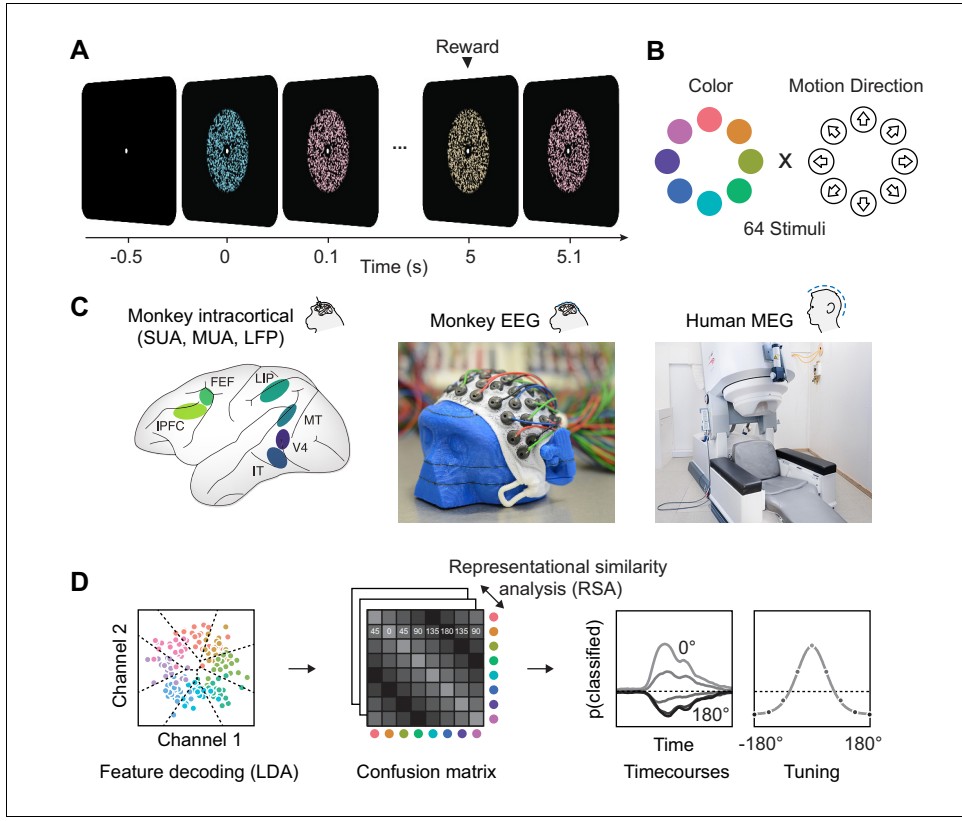

**Figure 1.** Experimental paradigm, recording and analyses. (**A**) We presented a stream of random dot patterns with rapidly changing colors and motion directions. After successfully maintaining fixation for a specified time, a liquid reward or auditory reward cue was delivered. (**B**) Colors and motion directions were sampled from geometrically identical circular spaces. Colors were uniformly distributed on a circle in L*C*h-space, such that they were equal in luminance and chromaticity, and only varied in hue. (**C**) We performed simultaneous microelectrode recordings from up to six cortical areas. We used custom 65 channel EEG-caps to record human-comparable EEG in macaque monkeys. MEG was recorded in human participants. (**D**) We used the same multivariate analysis approach on all signal types: Multi-class LDA applied to multiple recording channels resulted in time-resolved confusion matrices, from which we extracted classifier accuracy time courses and tuning profiles.
DOI: https://doi.org/10.7554/eLife.45645.003

confusion matrices indicate the average probability of the LDA to assign a trial of class A to class B. From this, we extracted information time courses, latencies, and tuning properties.

## Color and motion direction information in invasive and noninvasive signals

We found that information about both motion direction and color was present in all signal types (*Figure 2*). In LFP, multi-unit and single-unit data, motion and color information were strongest in areas MT and V4, respectively, in line with their established functional roles. Nonetheless, both features were represented ubiquitously (p<0.05, cluster permutation, for most areas apart from motion in IT LFP). Importantly, monkey EEG (*Figure 2B and E*) and human MEG (*Figure 2C and F*) also contained information about motion direction and color (p<0.05 for both features in both species, cluster-permutation).

Our analysis of microelectrode recordings showed decreasing information strength along the cortical hierarchy. To test whether this phenomenon was also detectable non-invasively, we performed source-reconstruction of monkey EEG and human MEG data using detailed physical headmodels (see methods, 'Source reconstruction and searchlight analysis'). We then repeated the MVPA in a searchlight fashion across the cortex. Indeed, for both monkey EEG and human MEG, this revealed gradients of information with strongest information in early visual areas (*Figure 2B,C,E,F*; insets).

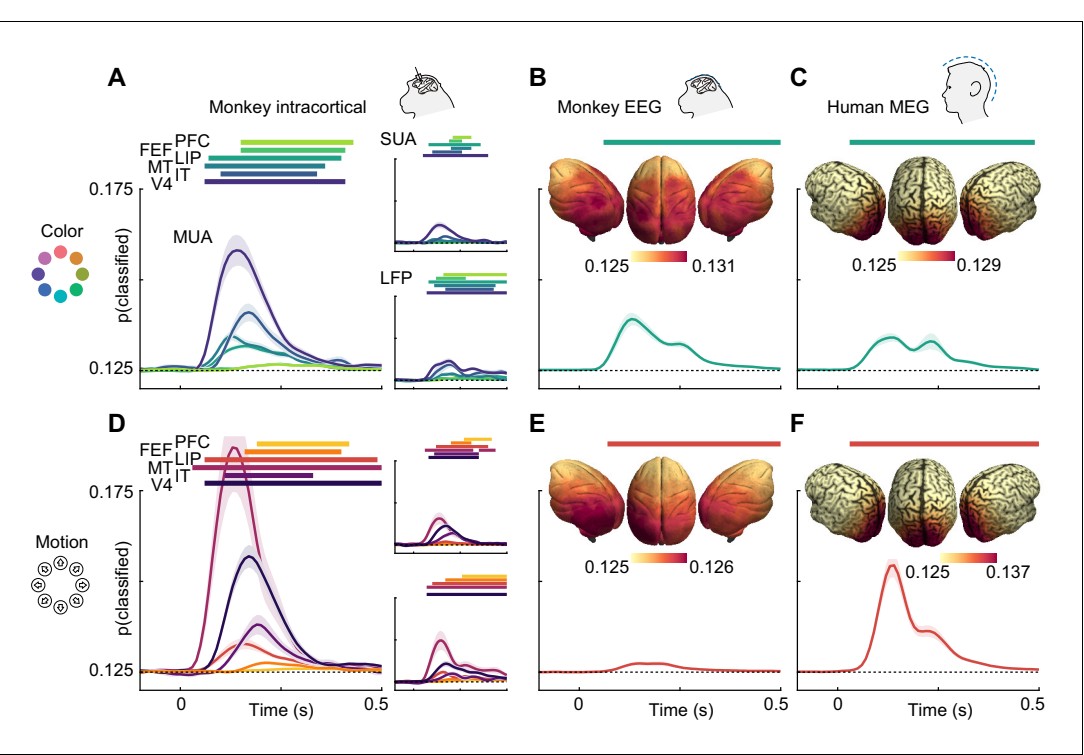

**Figure 2.** Color and motion direction information across areas and measurement scales. All panels show classifier accuracy, quantified as the single trial prediction probability for the correct stimulus. Error bars indicate standard error over recording sessions (in macaques) or participants (in humans), horizontal lines show periods of significant information (cluster permutation with p<0.05, corrected for number of regions). (A) Color and (D) motion information is available in most areas in multi-unit, single-unit and LFP data. Information decreases along the cortical hierarchy. Note that MUA color information has very similar timecourses in PFC and FEF, and thus, FEF is barely visible. (B) Color and (E) motion information is available in monkey EEG. Insets: distribution of information in monkey EEG, estimated using source-level searchlight decoding. Information peaks in occipital areas. (C) Color and (F) motion information is available in human MEG. Insets: distribution of information in human MEG, estimated using source-level searchlight decoding. Information peaks in occipital areas.
DOI: https://doi.org/10.7554/eLife.45645.004

To compare the dynamics of feature information, we estimated information latencies as the time point the decoding performance reached half its maximum (*Figure 3*). For the invasive recordings, latencies were in accordance with the visual processing hierarchy, with information rising earliest in MT for motion direction (SUA: 78 ms, MUA: 81 ms, LFP: 98 ms), earliest in V4 for color (SUA: 82 ms, MUA: 86 ms, LFP: 91 ms), and last in frontal areas. Generally, color information was available earlier than motion direction information in most areas where latencies could be estimated reliably for SUA (V4: p=0.001; IT: p=0.13; MT: p=0.39, random permutation), MUA (V4: p<0.001; IT: p=0.12; MT: p=0.26, random permutation) and LFP (V4: p=0.006; IT: p=0.75; MT: p=0.37, random permutation), consistent with previous results from the same animals in a different task (*Siegel et al., 2015*). These results translated to the noninvasive signals: Both for monkey EEG (color: 91 ms, motion: 103 ms, p=0.03, random permutation) and human MEG (color: 70 ms, motion: 97 ms, p<0.001, random permutation), color information rose earlier, and the latencies were comparable with those found invasively. Using the searchlight decoding analysis, we again found gradients consistent with the cortical hierarchy, with lowest latencies in occipital and highest latencies in more frontal regions (*Figure 3B, C,E,F*; insets), as confirmed by correlating source position and estimated latencies (MEG color: p=$10^{-15}$, MEG motion direction: p=$10^{-4}$, monkey EEG color: p=0.017, monkey EEG motion direction: p=$10^{-20}$).

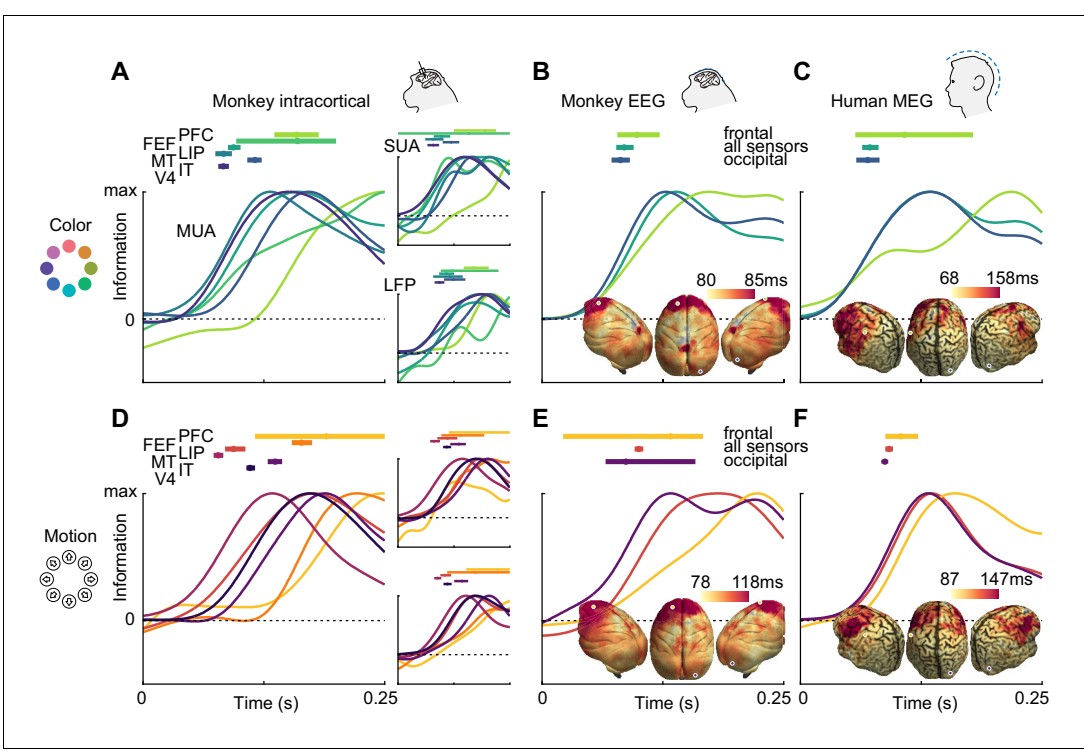

**Figure 3.** Color and motion direction information latencies across areas and measurement scales. All panels show normalized classifier accuracy, and latency estimates as well as confidence intervals (bootstrap, 95%). (**A**) Color and (**D**) motion information rises first in early visual areas, and last in frontal areas. (**B**) Color and (**E**) motion latencies in monkey EEG are comparable to those in early visual areas. Insets: distribution of latencies in monkey EEG, estimated using source-level searchlight decoding. Information rises later in frontal sources than in occipital sources. Marked positions indicate sources for which time courses are shown. (**C**) Color and (**F**) motion latencies in human MEG are comparable to those of early visual areas in the macaque brain. Insets: distribution of latencies in human MEG, estimated using source-level searchlight decoding. Information rises later in frontal sources than in occipital sources.
DOI: https://doi.org/10.7554/eLife.45645.005

## MEG color information cannot be explained by luminance confounds

Is it plausible that the contents of sensory representations are accessible to noninvasive electrophysiology? It has been shown that, in general, features represented at the level of cortical columns can propagate to decodable MEG and EEG signals (*Cichy et al., 2015*). Recently, it was reported that information about the motion direction of random dot stimuli can be extracted from EEG signals (*Bae and Luck, 2019*). This study is, however, to our knowledge the first direct report of color decoding from MEG or EEG. It is conceivable that luminance confounds introduced by imperfections in the color calibration or individual variation in retinal color processing could explain color decoding. To exclude this possibility, we performed a control experiment in a single human subject, in which we manipulated luminance such that each stimulus was presented in a darker and a brighter version. We then used a cross-classification approach to test whether true color information dominated the artificially introduced luminance effect. To this end, we grouped trials such that, for each color, one luminance level was used for training and the other for evaluating the decoder, effectively creating a mismatch of information between test and training data. The color decoder could now, in principle, pick up three sources of information: true color differences, unknown, confounding luminance differences, and experimentally introduced luminance differences. In isolation, these luminance differences should lead to below-chance accuracy. Therefore, any remaining above-chance effect would either indicate that the luminance confound was even stronger than the control manipulation, or that true color information was present. Indeed, we found that classifier accuracy was still significantly above chance ($p<0.05$, cluster permutation), and undiminished by the luminance manipulation (*Figure 4A*). Furthermore, we compared the confusion matrices of classifiers trained and tested on dark or bright stimuli, trained on dark and tested on bright stimuli, or vice versa (*Figure 4B*). All confusion matrices were highly similar, indicating that the representational structure was comparable for low- and high luminance colors. Taken together, this suggests that in our main experiment, equiluminant and equisaturated color stimuli lead to discriminable MEG signatures, and luminance confounds had only a small, if any, effect.

Our stimuli were generated in L*C*h-space, which is designed based on perceptual uniformity in humans. However, it has been shown that color sensitivities in macaque monkeys are highly similar,

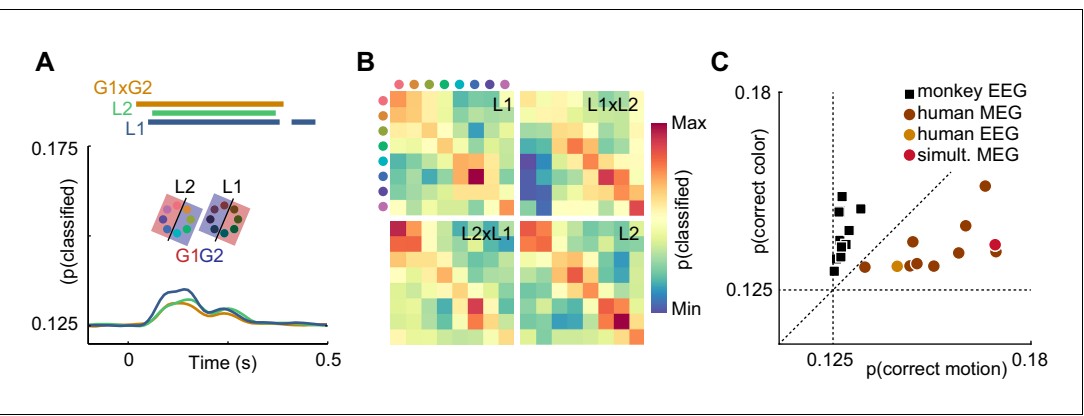

**Figure 4.** Control experiments. (**A**) Time-resolved classifier accuracy. Accuracy is highest when trained and tested on high-luminance stimuli (L1), and lower when trained and tested on low-luminance stimuli (L2). Training on half of the color space in low luminance, and half of the color space in high luminance, and testing on the remainder (G1 x G2), results in accuracy comparable to the low-luminance stimuli alone. (**B**) Confusion matrices for low-luminance and high luminance stimuli, as well as classifiers trained on low-luminance and tested on high-luminance stimuli and vice versa (L1 x L2, L2 x L1). (**C**) Maximum color and motion classifier accuracies for all individual sessions. Color is better classified in monkey EEG, motion is better classified in human MEG and EEG. Simultaneously recorded human MEG and EEG results in overall higher accuracy in MEG, but more motion than color information in both.

DOI: https://doi.org/10.7554/eLife.45645.006

The following figure supplement is available for figure 4:

**Figure supplement 1.** Perceptual equiluminance control.
DOI: https://doi.org/10.7554/eLife.45645.007

but not identical to humans (*Gagin et al., 2014*; *Lindbloom-Brown et al., 2014*). To ensure that color decoding in the monkey data was not driven by luminance differences, we performed a psychophysical control experiment in a third macaque monkey. Using a minimum-motion technique and eye movements as readout (*Logothetis and Charles, 1990*), we found that equiluminant colors generated in L*C*h-space were also close to perceptually equiluminant for this monkey (*Figure 4—figure supplement 1*).

## Information contained in human EEG is comparable to MEG

While in human MEG data, there was more information about motion direction than about color, monkey EEG data showed the opposite effect (*Figure 2B,C,E,F*). In principle, this could be due to differences in species, measurement modality (EEG or MEG), or differences in the visual stimulation that were beyond our control due to the separate recording environments. To exclude measurement modality as the relevant factor, we acquired simultaneous MEG and EEG data in one of the human participants and compared the amount of motion direction and color information across MEG and EEG data. All monkey EEG recording sessions contained more information about color, and all human MEG recordings contained more information about motion direction. Notably, the human EEG session was consistent with the MEG results. While information was generally lower for EEG than for simultaneous MEG, EEG showed the same dominance of motion information (*Figure 4C*). This suggests that the differences of information dominance between human MEG and monkey EEG were not due to the recording modality.

## Representational similarity analysis

Having established the presence of information in all signal types, we next asked how the representational structure of motion direction and color varied across brain areas, species, and measurement scales. To address this, we performed representational similarity analysis (*Kriegeskorte et al., 2008a*) (RSA) on the LDA confusion matrices averaged over a time window during which visual information was present (50–250 ms). In short, we used RSA to compare patterns of similarity between stimulus classes, as given by the confusion matrices, across areas and signal types. First, we sought to characterize the diversity of representations across the six areas measured invasively (*Figure 5A*). For color information, we found that representations were highly similar between SUA, MUA and LFP, as well as between all six cortical areas (p<0.05 for most pairs of areas and measures, uncorrected), indicating that a single representational structure was dominant across the brain. In the case of motion direction, areas were split into ventral stream visual areas (IT and V4) and frontal and dorsal visual stream areas (MT, LIP, FEF, PFC). Within each of these two groups, there were again high correlations between areas and measures, but we found no significant similarity between the groups.

How does information contained in locally recorded neuronal activity relate to information in large-scale EEG signals? We found that the color representation in macaque EEG was highly similar to those of SUA, MUA and LFP in all six areas, while the EEG motion direction representation reflected only the ventral stream areas V4 and IT (*Figure 5B*, p<0.05 for IT SUA and MUA, V4 SUA, MUA and LFP, random permutation, corrected for number of areas). Notably, we found no motion direction similarity between area MT and EEG (SUA: p=0.84; MUA: p=0.85; LFP: p=0.82, uncorrected). This implies that, although MT contained a large amount of motion direction information, EEG signals were dominated by activity from areas with V4- or IT-like motion direction tuning. We found similar results when comparing invasive data to human MEG; again, there were strong similarities between color representations in all areas and human MEG, as well as between motion direction representations in V4 and IT and human MEG (*Figure 5C*, p<0.05). Furthermore, both color and motion direction representations were highly similar between monkey EEG and human MEG (*Figure 5D*, color: r = 0.83, p=0.0002; motion: r = 0.69, p=0.0003).

## Similarity is explained by tuning properties

Color representations were similar across the brain, while motion direction representations were divided into two categories, only one of which translated to non-invasive signals. To investigate what led to these effects, we examined the underlying representations more closely. *Figure 6* shows the color and motion direction confusion matrices for MT and V4 multi-unit activity as well as for monkey

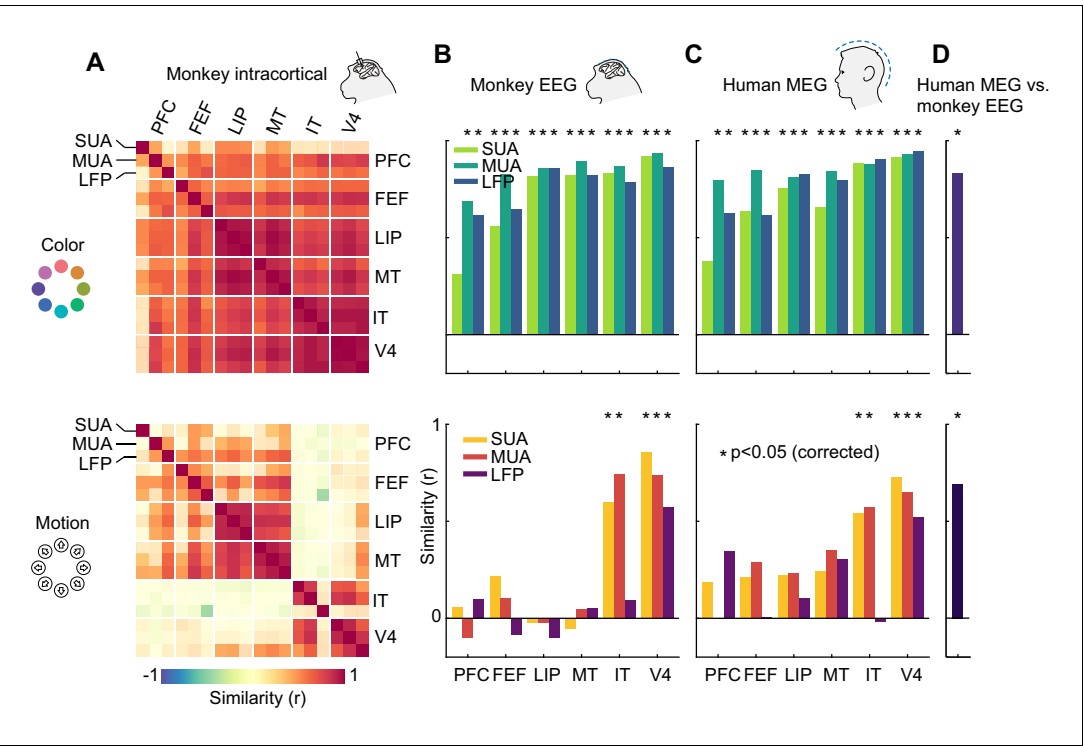

**Figure 5.** Representational similarity between areas and measurement scales. (**A**) Similarity between SUA, MUA and LFP color (top) and motion direction representations (bottom) in six areas of the macaque brain, masked at p<0.05 (uncorrected). Color representations are highly similar between all areas; motion representations are split between frontal/dorsal and ventral areas. (**B**) and (**C**) Similarity between monkey EEG and human MEG color and motion representations and those in SUA, MUA and LFP in six areas. Non-invasive color representations are similar to all areas, motion representations are similar to IT and V4 representations (p<0.05, random permutation test, corrected for number of areas). (**D**) Color and motion representations are similar between human MEG and monkey EEG (both p<0.001, random permutation test).

DOI: https://doi.org/10.7554/eLife.45645.008

EEG and human MEG. All color confusion matrices displayed a simple pattern decreasing with distance from the diagonal. This implies that neural activity distances in all areas, signals and both species approximately matched perceptual distances in color space. We found a similar representation of motion direction in area MT.

However, motion direction representations in V4, monkey EEG and human MEG displayed a distinct peak in similarity on the off-diagonal opposite to the true motion direction, indicating that these signals were, to some extent, invariant to motion in opposite directions. To assess the temporal dynamics of this effect, we collapsed the confusion matrices over stimuli, which results in prediction probabilities as a function of the angular difference between true and predicted stimuli (*Figure 6*). Here, the off-diagonal elements in the confusion matrices translated to an increased probability of a stimulus to be predicted as the one opposite in stimulus space. At all timepoints, color stimuli were least likely to be classified as the opposite color, whereas there was an increased probability for motion directions to be identified as the opposite. In terms of population tuning, this corresponds to bimodal tuning curves (*Figure 6*). We quantified the presence of such bimodal tuning across areas and measurement scales by calculating the slope in prediction probability between opposite (180-degree difference) and next-to-opposite (135- and 225-degree difference) stimuli, normalized by the range of prediction probabilities (*Figure 7*). This revealed that motion direction tuning was indeed significantly bimodal in V4 and IT as well as monkey EEG and human MEG, but not for any of the more dorsal or frontal areas. There was no significant bimodal color tuning for any area or measurement scale.

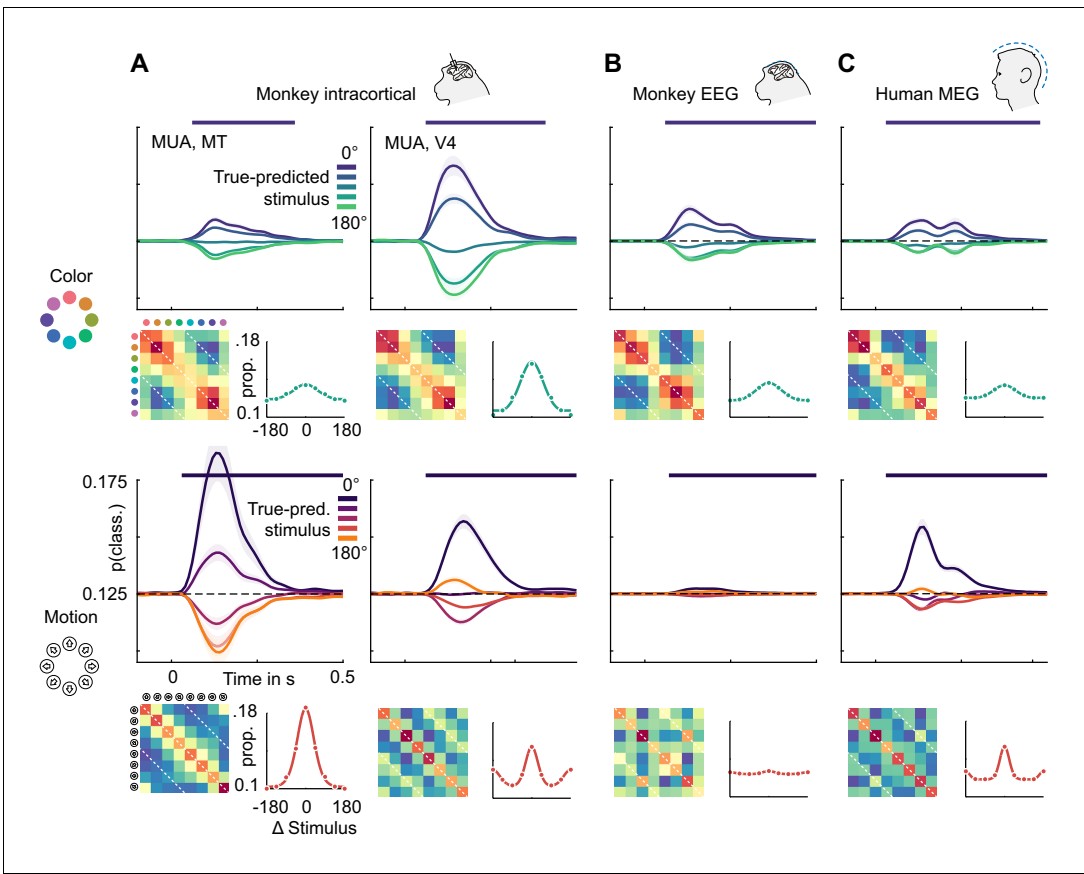

**Figure 6.** Color and motion direction tuning. Color and motion direction tuning in (**A**) MT (left), V4 (right), (**B**) monkey EEG and (**C**) human MEG. Shown is, for each area or signal type, first: the temporally resolved prediction probability as a function of the distance between true and predicted stimulus. The dark blue line indicates the probability of a stimulus being predicted correctly (classifier accuracy), the green (color) and orange (motion) lines the probability of a stimulus being predicted as the opposite in the circular stimulus space. Second: the confusion matrix, indicating prediction probabilities for all stimulus combinations. Third: A representation tuning curve, indicating prediction probabilities as a function of distance between true and predicted stimulus at the time of peak accuracy. For color, tuning is always unimodal, with opposite-classifications having the lowest probability. For motion direction, V4, EEG and MEG, but not MT tuning is bimodal, with opposite-classifications having a higher probability than some intermediate stimulus distances.
DOI: https://doi.org/10.7554/eLife.45645.009

We used linear regression to estimate the contribution of bimodality differences to the pattern of similarity between invasively measured areas and signal types (*Figure 7C and E*). To this end, we computed differences in bimodality between each combination of SUA, MUA and LFP, and all areas. We then assessed to what extent these differences in bimodality accounted for the variance in representational similarity. Importantly, in the case of motion direction, bimodality could largely explain the pattern of representational similarity between areas and measures ($R^2 = 0.28$, p=0). This was not the case for the small bimodality differences in color tuning, which did not affect representational similarity ($R^2 = 0$, p=0.99). Thus, similar motion direction bimodality led to V4 and IT showing similar motion representations, which were also similar to those in monkey EEG and human MEG.

## Motion direction bimodality is present in individual SUA, MUA and LFP channels

Finally, we asked on which level the motion direction bimodality arose. The presence of a bimodality effect in MEG, EEG, LFP, multi-unit and sorted unit data suggests that it was not caused by anisotropies in the large-scale organization of motion direction tuning, but rather by properties of individual

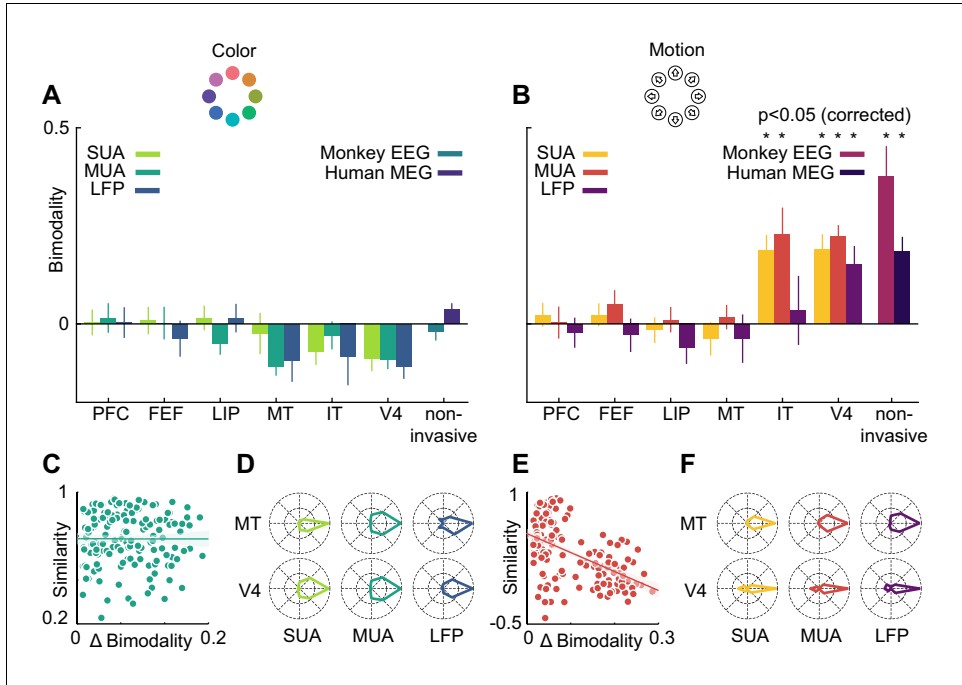

**Figure 7.** Motion direction tuning bimodality across areas and measurement scales explains representational similarity. Representations of color (A) do not show bimodality; representations of motion (B) do in IT, V4 and non-invasive data, but not in frontal and dorsal areas. (C, E). Correlation of representational similarity between SUA, MUA and LFP in all areas with differences in bimodality. In case of color (C), there is no strong correlation, in case of motion (E), there is a strong anticorrelation. (D, F). Average tuning of individual single units, multi units or LFP channels in MT and V4. Color tuning is unimodal in both areas, motion tuning is bimodal in area V4.
DOI: https://doi.org/10.7554/eLife.45645.010

units: if individual single or multi-units, or LFP channels, were able to distinguish between opposite motion directions, a multivariate analysis of several channels would be expected to also reflect this separability. We therefore expected bimodal motion direction tuning curves to be prominent in those areas which exhibited a multivariate bimodality effect. To test this, we aligned all tuning curves in V4 and MT according to their preferred direction and calculated, for each area, their average. Indeed, direction tuning curves in areas V4 (SUA: $p=1*10^{-9}$, MUA: $p=8*10^{-11}$, LFP: $p=0.03$) were bimodal, whereas direction tuning curves in area MT (SUA: $p=0.66$, MUA: $p=0.31$, LFP: $p=0.87$) or color tuning curves in either area (all $p>0.42$) were not (*Figure 7D and F*).

## Discussion

We found that information about motion direction and color was present in invasively recorded signals in many cortical areas in macaque monkeys as well as in non-invasive electrophysiological data from macaques and humans. Dissecting the information structure revealed representations according to perceptual similarity for color in all areas, and for motion direction in dorsal and frontal areas. Contrary to that, V4 and IT motion direction representations were bimodal, indicative of orientation rather than direction selectivity. We found the same bimodal pattern in monkey EEG and human MEG, as confirmed by representational similarity analysis. Together with converging evidence from latency and information distributions this pointed to early visual and ventral stream areas such as V4 as the main drivers of EEG and MEG representations, while dorsal areas like MT did not appear to strongly contribute to non-invasive signals.

### Widespread representations of visual features across cortex

Consistent with earlier reports (*An et al., 2012*; *Mendoza-Halliday et al., 2014*; *Siegel et al., 2015*), we found color and motion information in all areas we measured, rather than in a small

amount of specialized areas. Nonetheless, the amount of information strongly depended on the area. Interestingly, the motion direction decoding accuracies we found were lower than previously reported in both area MT and prefrontal cortex (*Mendoza-Halliday et al., 2014*). This can largely be attributed to differences in the paradigm and analysis strategy: First, rather than decoding from large pseudo-populations, we used small, simultaneously recorded populations. Second, we report averaged single trial probabilities, which tended to be smaller but more robust than the corresponding discrete classification results. Third, the rapid succession of very short stimuli likely limited cortical processing of each stimulus. Fourth, our paradigm only involved passive fixation. Especially in higher-order areas we would expect representations to be strengthened, and altered, according to task demands in a more engaging cognitive task.

## Early ventral stream areas as sources of non-invasive information

Stimulus features showing a spatial organization at the scale of cortical columns, such as orientation, can in principle be decoded from EEG and MEG (*Cichy et al., 2015*). This implies that other, similarly topographical representations should be equally accessible. Indeed, a clustering of both color (*Conway and Tsao, 2009*; *Roe et al., 2012*; *Tanigawa et al., 2010*) and motion direction (*Albright et al., 1984*; *Li et al., 2013*; *Tanigawa et al., 2010*) has been reported in several areas of the visual system. This suggests that our successful decoding of stimulus color and motion direction was not attributable to confounding factors, but likely stemmed from true feature-related signals.

Crucially, even though we recorded invasively in many areas, our results do not unequivocally identify the sources of visual information in MEG and EEG. First, neither color nor motion direction representations are limited to the areas we recorded from. Secondly, partially due to the simple feature spaces used for our stimuli, many areas are expected to show highly similar tuning properties. Based on RSA, we can therefore only conclude that the non-invasively measured information stems from areas with tuning similar to V4 or IT. It is reasonable to assume that earlier visual areas strongly contributed to this, which is corroborated by our source level searchlight analysis revealing strong information peaks in occipital cortex. Furthermore, it has been shown that for example area V1 exhibits a more bimodal motion direction tuning (i.e. orientation or axis tuning) than area MT (*Albright, 1984*), matching the results found here in V4. There is, however, previous evidence that the structure of color representations decodable from area V1 using fMRI is not in agreement with perceptual similarity (*Brouwer and Heeger, 2013*), contrary to area V4, and contrary to the representations we found in MEG and EEG, suggesting that these color representations might not be explained by V1 activity alone.

Notably, in area MT cortical columns with opposite preferred motion directions along the same axis lie spatially close to each other (*Albright et al., 1984*; *Born and Bradley, 2005*). This could, in principle, lead to a diminished decodability of opposite motion directions from mass signals such as EEG, MEG or fMRI. In such a scenario, the source of bimodal motion direction tuning might still lie in area MT. However, this would require columns with opposite preferred motion directions to be close to uniformly distributed at the relevant spatial scale. While several recent fMRI studies have focused on motion axis decoding (*Schneider et al., 2019*; *Zimmermann et al., 2011*), motion direction has been successfully decoded from BOLD signals in area MT (*Kamitani and Tong, 2006*). Given that motion representations are prevalent across visual cortex (*An et al., 2012*), we consider it unlikely that MT was a dominant source of the bimodally tuned motion signals we measured in EEG and MEG.

In sum, this suggests that the information decoded from non-invasive signals originated in a mixture of several early visual areas. Recordings from additional visual areas using the same paradigm are required to further clarify this. Future studies may also expand the stimulus space - a limitation of the present proof-of-principle study. Manipulating other stimulus features in order to maximize differences between areas will allow to further dissociate representations in specific parts of the visual system.

## Monkey EEG as a bridge technology

We utilized human-comparable monkey EEG as a bridge technology to link invasive animal electrophysiology to human MEG. High electrode density and methods identical to those used in human M/EEG enabled us to perform source reconstruction and directly relate measures across species.

The few available previous studies measuring EEG in nonhuman primates were typically restricted to only a few electrodes (*Bimbi et al., 2018*; *Snyder et al., 2015*; *Snyder et al., 2018*) and used diverging methods such as skull-screw electrodes, or both (*Godlove et al., 2011*; *Musall et al., 2014*; *Reinhart et al., 2012*; *Whittingstall and Logothetis, 2009*; *Woodman et al., 2007*). We show how monkey EEG can serve as a missing link to enable the disentangling of species differences from differences in measurement modality. In isolation, our observation of bimodal motion direction tuning in human MEG could not directly inform conclusions about the relative contributions of dorsal and ventral stream areas. Finding the same result in monkey EEG allowed us to infer that it was not due to a decreased influence of MT-like areas in the human, but rather a sign of a general dominance of V4-like tuning in non-invasive signals.

State-of-the-art animal electrophysiology requires large technical efforts and comes at a significant ethical cost. When applied in addition to ongoing invasive experiments, the marginal cost of monkey EEG is comparably small. It is non-invasive, depends mostly on standard human neuroscience tools, and does not necessitate further animal training. This is far outweighed by the potential benefits of establishing a database for linking invasive and non-invasive electrophysiology and for enhancing comparability between the fields. Notably, another possibility to achieve this goal is given by invasive electrophysiological recordings in human patients, that are however severely constrained by the requirement for medical necessity.

## A framework for linking measurement scales

In the current work, we used an information-based approach to compare brain areas, measurement scales, and species. Such analyses are powerful tools to relate very different signals based on their information contents. This may not only include data from different measurement techniques, such as MEG and fMRI (*Cichy et al., 2014*; *Cichy et al., 2016a*), or species (*Cichy et al., 2014*; *Kriegeskorte et al., 2008b*), but also cognitive or computational models (*Cichy et al., 2016b*; *Wardle et al., 2016*). Furthermore, instead of comparing representations of simple sensory stimuli, the same framework can be applied to complex task conditions (*Hebart et al., 2018*).

We would like to highlight that our framework of cross-species and cross-scale comparisons is not limited to information-based analyses. For example we anticipate that it will be highly interesting to compare and pinpoint specific spectral signatures of circuit activity in signals on all scales (*Donner and Siegel, 2011*; *Siegel et al., 2012*). This has been successful in some cases (*Sherman et al., 2016*; *Shin et al., 2017*), but could significantly benefit from the present large scale approach to gain further specificity. In the long term, with sufficient knowledge about mechanistic signatures on all scales, this could facilitate the establishment of transfer functions between circuit activity and non-invasive human electrophysiology (*Cohen, 2017*; *Donner and Siegel, 2011*; *Siegel et al., 2012*). It is important to note that such transfer can only be possible based on knowledge on all scales. As has been noted before (*Sprague et al., 2018*), macro-scale signals alone always suffer from an ill-posed inverse problem when trying to infer micro-scale properties. The approach of dense recordings on all scales, as outlined here, allows to bridge this gap by constraining inferences. Such developments would allow quick and inexpensive access to circuit function in the human brain, both for basic research and in clinical practice (*Siegel et al., 2012*).

## Summary and conclusion

In sum, we show that color and motion direction can be decoded from non-invasive electrophysiology in humans and monkeys. Our results suggest that such simple stimulus representations are dominated by signals from early ventral stream areas. This inference serves as a proof-of-principle for, and was enabled by, using high-density monkey EEG as a bridge technology to link scales and species.

## Materials and methods

### Macaque microelectrode recordings
#### Subjects
Microelectrode recordings were performed in two adult rhesus macaques, one male (monkey R) and one female (monkey P). Each monkey was implanted with a titanium headpost to immobilize the

head. Following behavioral training, three titanium recording chambers were stereotactically implanted over frontal, parietal, and occipitotemporal cortices in the left hemisphere. All procedures followed the guidelines of the Massachusetts Institute of Technology Committee on Animal Care and the National Institutes of Health.

## Stimuli and apparatus

We presented rapid streams of colored random dot kinematograms with 100% motion and color coherence. Colors and motion directions changed randomly from stimulus to stimulus. We sampled dot colors from a circle in CIEL*C*h color space such that they had equal luminance and saturation. The background color was always a uniform black. Therefore, individual stimuli contained both luminance and chromaticity contrasts between background and dots, whereas the only features varying over stimuli were color hue and motion direction. Sequences of stimuli were presented before each trial of an unrelated delayed saccade task and separated by short inter-stimulus intervals, while fixation had to be maintained. Stimuli had a diameter of 3.2 degrees of visual angle, featuring 400 dots with a diameter of 0.08 degrees. Two variants of this paradigm were used: in stimulus configuration A, we showed sequences of 6 stimuli lasting 150 ms with an ITI of 50 ms. In this case, 12 uniformly distributed colors and motion directions (0, 30, 60, 90, 120, 150, 180, 210, 240, 270, 300, 330 degrees) were used and dots moved at a speed of 10 degrees per second. In stimulus configuration B, sequences of 8 stimuli were shown. In this case, stimuli lasted 100 ms with ISIs of 20 ms, were sampled from eight colors and motion directions (0, 45, 90, 135, 180, 225, 270, 315 degrees), and dots moved at a speed of 1.33 degrees per second. Liquid rewards were administered when the monkeys succeeded in both maintaining fixation on the stimulus streams and in completing the subsequent unrelated trial. Stimuli were generated offline using MATLAB, and presented using the MonkeyLogic toolbox (*Asaad et al., 2013*).

## Microelectrode recordings

Microelectrode activity was recorded in a total of 71 recording sessions, 47 in monkey P and 24 in monkey R. 31 of the sessions in monkey P used stimulus configuration A, 16 used stimulus configuration B. 18 of the sessions in monkey R used stimulus configuration A, six used stimulus configuration B. Combined over all sessions of both monkeys, 58,056 stimuli were presented. In each recording session, we acutely lowered Epoxy-coated tungsten electrodes in up to six areas out of the lateral prefrontal cortex, frontal eye fields (FEF), lateral intraparietal cortex (LIP), inferotemporal cortex (TEO), visual area V4, and the middle temporal area (MT). Neuronal activity was recorded across a maximum of 108 electrodes simultaneously. All signals were recorded broad-band at 40 kHz referenced to the titanium headpost. Monkeys sat in a primate chair, while stimuli were presented on a CRT monitor with a refresh rate of 100 Hz.

## Preprocessing

We analyzed data from a total of 4505 cortical recording sites (V4: 372, IT: 148, MT: 272, LIP: 897, FEF: 1067, PFC: 1749). From the broad-band data, analog multi-unit activity (MUA) was extracted by high- and low-pass filtering at 500 and 6000 Hz, respectively (2nd-order zero-phase forward-reverse Butterworth filters), rectification, low-pass filtering at 250 Hz (2nd-order zero-phase forward-reverse Butterworth filter), and resampling at 1 kHz. Local field potentials (LFP) were extracted by low-pass filtering of broad-band data at 500 Hz and later re-referenced to a local bipolar reference. Single unit activity (SUA) was obtained through spike sorting (Plexon Offline Sorter) of the high- (500 Hz) and low- (6000 Hz) pass filtered broad-band data thresholded at four times the noise threshold. Single-unit isolation was assessed by an expert user (CvN) and judged according to a quality index ranging from 1 (clearly distinguishable, putative single unit) to 4 (clearly multi-unit). We used principal components (PC) 1 and 2 of the spike waveform as well as the nonlinear energy function of the spike as axes in 3D sorting space. A putative single unit had to exhibit clear separability of its cluster in this 3D feature space, as well as a clean stack of individual waveforms in its overlay plot. Units of all quality types were included in the analysis. All signal types were then band-pass-filtered between 0.1 and 10 Hz (Butterworth, 2-pass, 4th order). This transformed single unit spikes into an approximation of instantaneous firing rate and ensured comparability of all signal types with EEG and MEG data.

## Macaque EEG

### Subjects

We measured scalp EEG in two male adult rhesus monkeys. All procedures were approved by local authorities (Regierungspräsidium Tübingen).

### Stimuli and apparatus

Stimuli were created as described above for the macaque microelectrode recordings. However, we only used eight colors and motion directions, and no additional, unrelated task was performed. Initially, a central spot had to be fixated for 500 ms, after which stimuli started to appear for 100 ms each and without an ISI. Monkey V received a liquid reward as well as auditory feedback after 2 s of successful fixation on the stimulus sequence, after which the trial ended. For monkey E, we presented a continuous stimulus sequence as long as fixation was maintained. After each multiple of 5 s of successful fixation, reward and auditory feedback were administered. As soon as fixation was broken, the stimulus sequence stopped.

To maximize signal-to-noise ratio, we chose larger stimuli for most recording sessions: In all 3 sessions of monkey E, and 4 out of 8 sessions of monkey V, stimuli had a diameter of 6 degrees of visual angle, with a 0.75-degree central annulus. They consisted of 1600 dots with 0.2-degree diameter moving at 10 degrees per second. In the remaining 4 sessions of monkey V, stimuli had a diameter of 3.2 degrees, and consisted of 400 dots with 0.08-degree radius, therefore matching those used in the microelectrode recordings. Stimuli were generated offline using MATLAB and presented using Psychtoolbox (*Brainard, 1997*).

### EEG recordings

EEG was recorded using 65 Ag/AgCl electrodes and a NeurOne recording system (Bittium, Oulu, Finland) in 11 recording sessions, during which a total of 167,762 stimuli were presented. All channels were referenced to a central electrode, recorded with a sampling rate of 5 kHz and low-pass filtered online at 1250 Hz. An additional ground electrode was placed at the headpost. Electrodes were placed on the scalp using a custom-built 66-channel EEG cap (Easycap, Herrsching, Germany) covering the entire scalp. To leave room for the headpost, one of the 66 electrode positions was not used. Based on anatomical MRIs and 3D-printed head models, EEG caps were fabricated to match the individual animal's head shape. To achieve low impedances, we shaved and cleaned the monkeys' heads with saline and alcohol before each experimental session. Electrodes were filled in advance with a sticky conductive gel (Ten20, Weaver and Company, Aurora, Colorado, USA). After placing the cap on the head, we applied a second, abrasive conductive gel (Abralyt 2000, Easycap, Herrsching, Germany) through the opening of the electrodes, yielding stable impedances below 10 kΩ. Before each session, we 3D-localized electrode positions relative to the head using a Polaris Vicra optical tracking system (NDI, Waterloo, Ontario, Canada). Monkeys sat in a primate chair in a dark recording chamber while stimuli were presented on a CRT monitor with a refresh rate of 100 Hz. Infrared eye-tracking was performed at a sampling frequency of 1000 Hz using an Eyelink 1000 system (SR Research, Ottawa, Ontario, Canada).

### Preprocessing

EEG data was down-sampled to 300 Hz, re-referenced to an average reference and band-pass-filtered between 0.1 and 10 Hz (4th order, forward-reverse Butterworth filter).

## Human MEG

### Subjects

11 healthy volunteers (three female, 28.6 + −4.8 years) with normal or corrected-to-normal vision participated in this study. They received monetary rewards for participation that were in part dependent on their performance on the task. The study was conducted in accordance with the Declaration of Helsinki and was approved by the ethics committee of the University of Tübingen. All participants gave written informed consent before participating.

## Stimuli and apparatus

Stimuli were created and presented as described above for the monkey EEG recordings. Random dot kinematograms had a diameter of 6 degrees, with a central annulus of 0.75 degrees, and were presented in a continuous stream that ended when fixation was broken. After each multiple of 5 s of successful fixation, participants received auditory feedback associated with a monetary reward. Stimuli were generated offline using MATLAB, and presented using Psychtoolbox (*Brainard, 1997*).

## MEG recordings

We recorded MEG (Omega 2000, CTF Systems, Inc, Port Coquitlam, Canada) with 275 channels at a sampling rate of 2,343.75 Hz in a magnetically shielded chamber. The eleven participants completed one recording session each, resulting in a total of 237,348 stimuli being presented. Participants sat upright in a dark room, while stimuli were projected onto a screen at a viewing distance of 55 cm using an LCD projector (Sanyo PLC-XP41, Moriguchi, Japan) at 60 Hz refresh rate.

## Preprocessing

MEG data was downsampled to 300 Hz and band-pass-filtered between 0.1 and 10 Hz (4$^{th}$ order, forward-reverse Butterworth filter).

## Structural MRI

To enable source reconstruction, we acquired anatomical MRI scans from both macaques and humans. T1-weighted images were obtained for each human participant and the two monkeys used for EEG recordings.

## Multivariate classification

We used linear discriminant analysis (LDA) to extract the content and structure of information about stimulus features from all signal types. Trials were stratified such that each combination of color and motion direction occurred equally often, grouped according to one of the two stimulus features and split into training and test sets. For each time-point, we trained multi-class LDA on the training set, and predicted stimulus probabilities in the test set, using the activity in single or multi-units, EEG electrodes or MEG sensors as classification features. From the predicted stimulus probabilities, we created confusion matrices indicating the probability of stimuli being labeled as any other stimulus by the classifier. We evaluated classifier performance as the hit rate, calculated as the mean of the diagonal of the confusion matrix.

For EEG and MEG, we repeated this analysis in a 10-fold cross-validation scheme for each session, using all available sensors. For SUA, MUA and LFP, we used 2-fold instead of 10-fold cross-validation. Here, stimuli were presented in sequences of six or eight stimuli, and the occurrence of individual stimuli at each sequence position was not fully balanced. To prevent a potential confound of stimulus information with sequence position, we chose a stratification approach that kept the number of occurrences of each stimulus at each sequence position identical by oversampling the under-represented stimuli within each cross-validation fold. Due to the relatively low number of stimuli per recording session, 10-fold cross-validation would not have resulted in sufficient trials per fold for this approach. We therefore chose 2-fold cross-validation instead and performed classification independently for each of the six areas recorded. We restricted the analysis to five units per area at a time and repeated it for all or maximally 40 random combinations of the available units, to enable a comparison of information content in different areas. Results from these repetitions were averaged before statistical analysis. This analysis was performed for each time point from 250 ms before to 500 ms after stimulus onset, in steps of 10 ms, resulting in confusion matrices and classifier performances at 76 time points. In most of our recordings we presented eight different colors or motion directions. However, in the invasive recordings in stimulus configuration A there were 12 colors and directions. Therefore, we interpolated the confusion matrices of these recordings from a 12 × 12 to an 8 × 8 space.

We assessed the presence of significant information using a cluster sign permutation procedure (similar to *Cichy et al., 2014*). After subtracting chance performance (0.125), we determined temporally contiguous clusters during which information was higher than 0 (one-tailed t-test over recording sessions, p<0.01). We then randomly multiplied the information time-course of each recording

session 10,000 times with either 1 or −1, resulting in an expected value of 0. In each random permutation, we re-computed information clusters and determined the cluster-mass of the strongest cluster. Each original cluster was assigned a p-value by comparing its size to the distribution of sizes of the random permutation's strongest clusters.

## Latencies

Information latency was computed as the time point classifier performance reached half its peak. The peak was estimated as the first local maximum in performance that reached at least 75% of the global performance maximum. To avoid latencies being dominated by those recording sessions containing the most information, we normalized each session's classifier performance and used only those sessions where the post-stimulus performance peak was at least 1.5 times higher than the largest deviation during pre-stimulus baseline.

We estimated 95%-confidence intervals using bootstrapping. To statistically assess latency differences between color and motion direction, we used a random permutation procedure. True differences were compared to a distribution of latency differences generated from 10,000 random permutations of the group labels. To test whether latencies in the source-reconstructed monkey EEG and MEG systematically varied along the occipito-frontal gradient, we selected all sources containing significant information (cluster permutation, p<0.05). We then computed Pearson correlation coefficients between the physical location of those sources along the occipito-frontal gradient and the estimated latencies.

## Luminance control

To control for possible effects of luminance on color classification, we measured MEG as described above in one human participant during an additional control experiment. For this experiment, we used the same stimulus space as for the main experiment, but additionally included each color at a lower luminance level, such that the luminance contrast between colored dots and background was 20% lower. We then employed the same multivariate classification approach, but split training and test data according to their luminance levels. First, we used only either low-luminance or high-luminance trials for both training and testing. Second, we repeatedly split the color space into two halves, along each possible axis, trained on high-luminance stimuli from the one half and low-luminance stimuli from the other, and tested on the remaining stimuli. We then averaged confusion matrices over all axes, before extracting classification accuracies. To assess statistical significance, we repeated the analysis 100 times after shuffling the stimulus labels; the distribution of accuracies from shuffled data was used to compute p-values for the unshuffled data.

## Macaque equiluminance control

As color vision in macaques and humans is slightly different, we performed a psychophysical control experiment in a third macaque monkey to assess if our stimuli were in fact perceptually equiluminant to macaque monkeys. To this end, we used an adapted minimum motion technique using eye-movements as a readout (*Logothetis and Charles, 1990*). We measured small eye movements while the monkey was required to hold fixation on sequentially presented grating stimuli. Each stimulus lasted 500 ms and consisted of a repeating sequence of 4 frames, where frames 1 and 4 contained luminance contrast gratings, whereas frames 2 and 3 contained a contrast between a reference gray of a defined luminance and the probe color we wanted to assess. The phase of each grating proceeded by a quarter cycle with respect to the previous one, such that a probe color of higher luminance than the reference gray would elicit a motion percept in one direction, whereas a probe color of lower luminance would elicit a motion percept in the opposite direction. A probe color of the same luminance as the reference gray should not elicit any consistent perceived motion. Each stimulus was presented in two conditions: In the first condition, a color of higher luminance would elicit upwards motion, in the second one it would elicit downwards motion. We showed stimuli in trials of four, where subsequent stimuli always belonged to the opposite condition. We computed the difference in eye trace curvature – the second derivative of the vertical eye position over time - between conditions as a measure of perceived luminance deviation from the reference gray. We used this procedure for colors of eight hues in L*C*h-space, as in the main experiment. Stimuli of each color were generated at 19 L values, centered around the L value of the reference gray. The reference gray was

chosen as the center of the largest possible equiluminant circle in L*C*h-space, such that it was comparable in luminance to the stimuli used in the main experiment. Using linear regression, we assessed at which L value the luminance difference measure crossed 0, which established the point of perceptual equiluminance.

### Human EEG control

To assess whether the inverted relationship between color and motion information in monkeys and humans was due to differences between EEG and MEG, we simultaneously measured EEG and MEG in one of the eleven human participants. Identical analyses were performed on the human EEG data, and we compared maximal accuracies for color and motion decoding in all monkey EEG and human MEG sessions as well as the human EEG session.

### Source reconstruction and searchlight analysis

To assess the distribution of information in human and macaque brains, we performed source reconstruction on monkey EEG and human MEG data and repeated the multivariate classification in a searchlight fashion. We used structural MRI scans to create individual realistic head models. For MEG source reconstruction, we generated single-shell head models (*Nolte, 2003*). In the case of EEG source reconstruction, we manually extracted the skull and then segmented the brain into gray matter (GM), white matter (WM) and cerebrospinal fluid (CSF) using SPM and fieldtrip toolboxes in combination with probabilistic tissue maps for the macaque brain (*Rohlfing et al., 2012*). We determined the position of the titanium headposts with respect to the head surface using an optical tracking system, and incorporated 3D models of the headposts into our segmentation. These overall six tissue types (WM, GM, CSF, skull, scalp, titanium) were then used to generate detailed finite element head models (FEM) using the SimBio toolbox (*Fingberg et al., 2003*) as implemented in Fieldtrip. We estimated human MEG source activity at 457 and monkey EEG source activity at 517 equally spaced locations on the cortical surface, using linear spatial filtering (*Van Veen et al., 1997*).

### Representational similarity analysis

We compared representational structure between brain areas, measurement methods and species using representational similarity analysis. To this end, we computed the temporal average of the confusion matrices over a time period in which stimulus information was available (50–250 ms). Each entry in the resulting matrix gave an approximation of the similarity between stimulus representations. We then performed RSA by correlating matrices, after removing the diagonal. To assess significant similarity, we used a permutation procedure in which we randomly reassigned stimulus labels to the rows and columns of the confusion matrices 10,000 times. P-values were computed as the probability that the similarity between shuffled matrices deviated from zero at least as strongly as the true similarity.

### Population tuning properties

From the time-averaged confusion matrices, we extracted several tuning parameters to identify the factors contributing to similarity across scales and species. First, we collapsed confusion matrices across stimuli to obtain tuning curves denoting classifier prediction probability as a function of distance between stimuli. In these tuning curves, a peak at zero indicates a high probability of a stimulus being correctly identified by the classifier, and a peak at 180 degrees indicates an elevated probability of a stimulus being identified as its opposite. We estimated population tuning bimodality by computing the difference between opposite (180 degrees) and next-to-opposite (135, 225 degrees) stimuli normalized by the difference between maximal and minimal prediction probabilities. This bimodality-index is positive in case of a second peak at 180 degrees and zero or negative in case of a unimodal tuning curve. We used t-tests over sessions or subjects to test statistical significance of the bimodality (bimodality-index>0). To estimate the importance of bimodality for representational similarity, we computed the differences in bimodality between all invasively measured areas and signal types. We then used linear regression to determine the amount of variance in the representational similarities explained by these bimodality differences.

## Single channel tuning

To estimate average tuning curves of single units, multi-units and LFP channels in each cortical area, we performed one-way ANOVAs on each channel to select those containing information about color or motion direction, respectively, with a statistical threshold of $p < 0.05$. We then computed single-channel tuning curves and aligned them according to their preferred stimulus, determined as the stimulus for which firing rate, or LFP power, was highest. Finally, we computed the mean of all aligned tuning curves within one area, for each signal type. To assess single-unit bimodality, in a given area, we used one-sided t-tests to assess if the above described bimodality index was larger than 0.

## Software

All analyses were performed in MATLAB, using custom code as well as the Fieldtrip (*Oostenveld et al., 2011*) and SPM toolboxes.

## Acknowledgements

The authors thank Nima Noury for help with the monkey EEG recordings.

## Additional information

### Funding

| Funder | Grant reference number | Author |
|---|---|---|
| National Institute of Mental Health | R37MH087027 | Earl K Miller |
| European Research Council | StG335880 | Markus Siegel |
| Deutsche Forschungsgemeinschaft | 276693517 (SFB 1233) | Markus Siegel |
| Deutsche Forschungsgemeinschaft | SI1332-3/1 | Markus Siegel |
| Centre for Integrative Neuroscience | EXC 307 (DFG) | Markus Siegel |

The funders had no role in study design, data collection and interpretation, or the decision to submit the work for publication.

### Author contributions

Florian Sandhaeger, Conceptualization, Formal analysis, Investigation, Methodology, Writing—original draft; Constantin von Nicolai, Investigation, Methodology, Writing—review and editing; Earl K Miller, Resources, Funding acquisition; Markus Siegel, Conceptualization, Resources, Supervision, Funding acquisition, Investigation, Methodology, Writing—review and editing

### Author ORCIDs

Florian Sandhaeger (ID) https://orcid.org/0000-0002-9633-9556
Constantin von Nicolai (ID) https://orcid.org/0000-0002-8928-5860
Markus Siegel (ID) https://orcid.org/0000-0001-5115-936X

### Ethics

Human subjects: The study was conducted in accordance with the Declaration of Helsinki and was approved by the ethics committee of the University of Tuebingen (615/2017BO2). All participants gave written informed consent before participating.
Animal experimentation: Microelectrode recordings were performed in two adult rhesus macaques. All procedures followed the guidelines of the Massachusetts Institute of Technology Committee on Animal Care and the National Institutes of Health. Scalp EEG was measured in two male adult rhesus

monkeys. All procedures were approved by local authorities (Regierungspräsidium Tübingen, CIN 3/14).

## Decision letter and Author response
Decision letter https://doi.org/10.7554/eLife.45645.015
Author response https://doi.org/10.7554/eLife.45645.016

# Additional files

## Supplementary files
• Transparent reporting form
DOI: https://doi.org/10.7554/eLife.45645.011

## Data availability
Data and MATLAB code required to reproduce all figures are available at https://osf.io/tuhsk/.

The following dataset was generated:

| Author(s) | Year | Dataset title | Dataset URL | Database and Identifier |
|---|---|---|---|---|
| Florian Sandhaeger, Constantin von Nicolai, Earl K Miller, Markus Siegel | 2019 | Monkey EEG links neuronal color and motion information across species and scales | https://dx.doi.org/10.17605/OSF.IO/TUHSK | Open Science Framework, 10.17605/OSF.IO/TUHSK |

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
