## [Decision Letter]

Thank you for submitting your article "Monkey EEG links neuronal color and motion information across species and scales" for consideration by *eLife*. Your article has been reviewed by two peer reviewers, and the evaluation has been overseen by a Reviewing Editor and Joshua Gold as the Senior Editor. The following individuals involved in review of your submission have agreed to reveal their identity: Klaus Wimmer (Reviewer #1); Roger Tootell (Reviewer #2).

The reviewers have discussed the reviews with one another and the Reviewing Editor has drafted this decision to help you prepare a revised submission.

Summary:

Your manuscript was well received by both reviewers. They each commented on the pioneering nature of the work comparing neural signals recorded at different spatial scales in humans and macaques and on its relevance to the wider neuroscience community. However, they raised a number of issues related to analysis and decoding of neuronal activity recorded in monkeys. In addition, there were reservations concerning the interpretation of the underlying neural selectivity and its sources. These issues must be addressed before the paper will be considered for publication in *eLife* and are listed below.

Essential revisions:

1) Please address the reasons for the low decoding accuracy of MT activity compared to the published reports

2) Please comment on the significance levels for the low accuracy decoding results. Please perform comparable cross validation for the neuronal recordings and for the EEG and MEG, as suggested by reviewer 1.

3) Please address the possibility that stimulus representation in higher brain regions may be weakened in passively fixating subjects.

4) Reviewer 2 pointed out that direction-selective activity is not limited to neurons residing in area MT, which in rhesus monkeys is a small and deep structure, compared to other possible sources of direction selective activity in visual cortex. Please address the possibility of other sources of direction selectivity in the EEG activity recorded in monkeys and MEG in humans.

5) Please correct the placement of the sampling site of area V4 in Figure 1C and the point that color selective signals are not confined to area V4.

6) Please provide a detailed description of the color stimulus.

7) In the Discussion, please address reviewer 2's comment on bimodality. This reviewer highlights a possibility that direction columns have a spacing much smaller than that for axis of motion, a possibility raised by the early columnar models in macaque (Albright et al., 1984; Born and Bradley, 2005). A possibility supported by high resolution fMRI data that the "axis of motion" columns in human MT/V5 may include neurons of opposite direction selectivity (Zimmerman et al., 2011; Schneider et al., 2019). If the direction columns are significantly smaller than those of axis of motion, the source localization and measurements of bimodality may differ from each other in MT and other areas. In this context, the work of An et al., 2012 pointing to sources of motion signals in cortical areas outside MT should also be considered.

8) In Figure 2, please change blue and green colors for different cortical areas, to make the easier to distinguish and improve the visibility of the two lines in Figure 2A.

9) Provide the missing details highlighted by reviewer 1.

Complete reviews for your information:

Reviewer #1:

This manuscript investigates the representation of color and motion direction in human and macaque cerebral cortex, using: (i) simultaneous microelectrode recordings from 6 cortical areas in the macaque, (ii) high-density EEG again in the macaque, and (iii) MEG and EEG in humans. It is argued that information about motion and color is present in all signal types and both species. The pattern of stimulus encoding in EEG and MEG signals was similar to microelectrode recordings in ventral visual areas but not dorsal or frontal areas, clarifying thus the origin of color and motion information in the non-invasive recordings.

The combination of invasive and non-invasive recordings in humans and in macaque is an experimental tour de force and relating these signals obtained at different spatial scales is of broad interest to the systems neuroscience community. This is pioneering work. I have read the manuscript with great interest. Relating the EEG and MEG signals to tuning properties of neurons in V4 and MT (direction vs. orientation-tuned) is a convincing example of the usefulness of this approach.

However, I think there are some limitations of the study which should be discussed, and I have also identified some issues which need clarification.

1) The decoding accuracy for single-unit and multi-unit activity is very low and this puzzles me. Let's take MT as an example: coding accuracy is well below 0.2 in all cases (chance level is at 0.125). We know that the majority of neurons in MT is tuned to motion direction and 100% coherent random dot stimuli are perhaps the stimuli that best drives these neurons. Thus, I expected a decoding accuracy for 1 out of 8 directions close to 1.0 in MT. In fact, Mendoza-Halliday et al., 2014 reported an accuracy > 0.5 even in PFC. The single unit tuning curves in Figure 7F also show high selectivity. What explains the low accuracy in the manuscript (e.g. Figure 2)? Is it because of the bandpass filtering of spike trains? I think in order to properly relate invasive and non-invasive recording it must be assured that both measurements are consistent with previous literature.

2) Statistical significance of decoding results. Most of the decoding accuracies shown in Figure 2 are really low, sometimes ~0.1255 to 0.126 when the chance level is 0.125. Frankly, I don't understand how such small effects can be significant given the number of subjects and trials. It is crucial to confirm that some of the tiny effects are not caused by confounds such as any small imbalance in the data that could be detected by the decoder (unequal trial numbers, differences between individuals, etc.). At the very least I would suggest running the whole decoding analysis on data with shuffled stimulus class labels. This should not yield significant decoding apart from a few false positives. More detailed questions concerning the statistical procedures: Why only a 2-fold cross validation for SUA and MUA and not 10-fold as for EEG and MEG? Cluster permutation procedure (subsection “Multivariate classification”, last paragraph): I do not understand the rationale of multiplying with +1 or -1. This should be explained better and a reference should be given. Finally, is the use of t-tests appropriate here?

3) One limitation of the study is that subjects view the random dot patterns just passively. This may reduce the stimulus representation in higher brain areas (see e.g. work of Romo and Pasternak with active and passive tasks). This should be discussed.

4) Data availability. The authors have put the data points that are necessary to plot the figures in a repository, but this is of limited use for the community. I would strongly suggest making the full data available for further analysis. I hope this request is consistent with the spirit of *eLife*. The main advance of this manuscript is methodological, and I believe that providing this data as a resource to the community would be a further argument for publishing it in *eLife*.

Reviewer #2:

General:

The overall goal of synthesizing work from humans and macaques, and across different functional tools (e.g. EEG, electrophysiology and MEG), and across spatial scales, is very important. In that sense alone, this study is valuable and should have impact. Also, the analysis is relatively sophisticated and quantitative. However, the biological interpretation of the underlying neural selectivity is a little naïve, which weakens support for the overall conclusions. Presumably, the authors could mitigate the latter concern by revising the text accordingly.

Specific:

Based on the wealth of prior research in both macaques and humans, it is well accepted that most/many neurons in area MT/V5 respond to stimuli in a direction-selective manner. However it is important to remember that area MT/V5 is a relatively small area, and not located on the cortical surface (thus arguably less amenable to accurate source analysis). More importantly, many other cortical areas (in areas V1, V2, V3, V3A, MSTd/l, etc.) also have many direction-selective neurons (or direction-activated) neurons, and most of those areas are much larger compared to MT/V5. Thus it seems likely that such areas contribute significantly to the 'motion' contrast tested here; i.e. it was not produced entirely (or perhaps even preferentially) in area MT/V5.

The attribution of color-selective responses to 'V4' is less certain. For one thing, the oval indicating the 'V4' sampling site in macaque (Figure 1C) is overlaid on foveal V1/V2, not V4. The confusion is partly due to the presence of a very small gyrus (between the operculum and lunate gyrus) that connects the foveal representations of V1, V2, V3 and V4. A second concern is that color selective neurons and columns/patches/responses are not confined to (nor prominently present in) area V4; that is a persistent myth based on a few recordings made in the 1970s. Subsequent research has demonstrated many color responsive columns/patches/response in many other areas, (although not in MT). Thirdly, the color stimulus in 1A appears to be a mix of black dots and colored dots – i.e. a stimulus which varies in both luminance and color. [However I could not find a direct description of the color stimulus – so I am not certain about the stimulus]. Fourth, the color selectivity in humans (e.g. the L*C*h* space) differs from that in macaques, partly because the ratio of long- to medium-wavelength cones varies by a factor of two between these two species.

Thus, I conclude that the measured responses to these different stimuli may well differ as reported in the recording sites, consistent with the authors interpretation. However, the simple attribution of color-vs.-motion selectivity to MT and V4 needs to be significantly qualified. Presumably if the authors had been able to directly map the cortical sites that respond selectively to color or motion everywhere in visual cortex, the source localization would have been more certain.

The source localization of MEG and EEG information is well known to be roughly inverted, differentially maximal from sulci vs. gyri in the two measurements. This issue is crucial in the determination of source localization in brain. The authors might explore and discuss this in more detail; here I instead got the impression that the EEG and MEG signals were similar.

The extensive discussion of bi-modality (vs. lack thereof) in color vs. motion may change during the revision. However, it may be relevant that more recent attempts to map 'direction' columns (i.e. a dimension of 360 degrees) in cortex have yielded only axis-of-motion columns (a 180 degree dimension). Also, as briefly discussed by the authors, attempts to model color-selective cortical responses as psychophysically-color-opponent signals have not been entirely successful. These backgrounds may clarify the bi-modality discussion.

---

## [Author Response]

Essential revisions:1) Please address the reasons for the low decoding accuracy of MT activity compared to the published reports.

We thank the reviewers for bringing this to our attention. We added a paragraph discussing the availability and magnitude of information across cortical areas, with a focus on MT and PFC, to the Discussion section (subsection “Widespread representations of visual features across cortex”).

Briefly, there are several methodological differences that we believe account for the lower decoding accuracies in MT and PFC than in Mendoza-Halliday et. Al., 2014. While they decode from larger pseudo-populations of 30 not simultaneously recorded neurons, we decode from populations of 5 simultaneously recorded signals. Furthermore, our paradigm consisted of a rapid succession of very short, only passively fixated stimuli, which arguably does not maximize decodability. Lastly, we report the average of single-trial LDA probabilities instead of discrete classification results. When using discrete classifier outputs, the information in each trial is reduced to which of the class distributions it lies closest to, whereas LDA probabilities retain continuous information about the relative distances to all classes. This made our information estimates more robust, but decreased decoding accuracies.

2) Please comment on the significance levels for the low accuracy decoding results. Please perform comparable cross validation for the neuronal recordings and for the EEG and MEG, as suggested by reviewer 1.

We thank the reviewers for this comment, which made us realize that we did not sufficiently describe our statistical procedures. As suggested by reviewer 1, we expanded on our explanation of the sign permutation test in the Materials and methods section and added a reference for a paper using a similar test in a similar context.

Such a test (performed on the session level) can result in statistical significance for low decoding accuracies, if these are consistently above chance across recording sessions. To double-check the results, we performed the identical analysis on trial-shuffled data, as suggested by reviewer 1, which revealed – as expected – no significant clusters for any combination of visual feature, area, and measurement method. Thus, there are no false positive results. The results of this control analysis are shown in Author response image 1.

Furthermore, we added an explanation of the use of the 2-fold cross-validation procedure in the invasive data to the Materials and methods section (subsection “Multivariate classification”, second paragraph). Briefly, this was done to enable the control of a potential confounding factor: Stimuli were presented in groups of 6 or 8 at the beginning of each trial of another task, but different stimuli were not fully balanced over sequential positions within these groups. As this could have led to a confound of stimulus information with sequence position, we chose to use a stratification approach in which the number of occurrences of each individual stimulus was kept the same for all sequence positions in each cross-validation fold. Due to the overall low number of trials, 10-fold cross-validation did not result in sufficient trials per fold for this approach. However, due to the larger number of recording sessions as compared to the non-invasive data, we could still arrive at robust information estimates with 2-fold cross-validation, and the essential logic of independent training and test sets was not affected by using 2 instead of 10 folds.

We agree with the reviewer that to achieve maximal comparability, the identical analysis as performed in the non-invasive data is of interest. We therefore repeated the decoding of aMUA, SUA and LFP data using a 10-fold cross-validation, without the serial position stratification, with highly similar results in terms of overall decoding accuracy and statistical significance, with the exception of occasional minimal amounts of baseline information (see e.g. V4 LFP) which we assume to be the result of the confound described above. The results of this control analysis are attached as Author response image 2.

**Author response image 2. respfig2:** 

3) Please address the possibility that stimulus representation in higher brain regions may be weakened in passively fixating subjects.

We thank the reviewers for this comment, that we fully agree with. We have incorporated a short consideration into the same paragraph discussing decoding strength in different cortical areas (see 2., and subsection “Widespread representations of visual features across cortex”).

4) Reviewer 2 pointed out that direction-selective activity is not limited to neurons residing in area MT, which in rhesus monkeys is a small and deep structure, compared to other possible sources of direction selective activity in visual cortex. Please address the possibility of other sources of direction selectivity in the EEG activity recorded in monkeys and MEG in humans.

We fully agree with the reviewer that, also because of the size and positioning of MT, the motion direction information in EEG/MEG was likely also influenced by other areas than MT. To clarify this, we have adapted parts of the Discussion section: first, we state more clearly now that not only the areas that we recorded from, but also many other areas are selective for both motion and color and could therefore contribute to EEG/MEG decoding. Secondly, (mainly in response to point 7 below), we have expanded on the consequences of anatomical organization within MT (subsection “Early ventral stream areas as sources of non-invasive information”).

5) Please correct the placement of the sampling site of area V4 in Figure 1C and the point that color selective signals are not confined to area V4.

We have corrected the placement of V4 in the schematic and clarified in the Discussion that both color and motion information was found in all areas we recorded from, but is also expected in many other areas (see also response to point 4 above).

6) Please provide a detailed description of the color stimulus.

We added the missing information about the stimulus background color to our description of the stimuli in the Materials and methods section / Macaque microelectrode recordings / Stimuli and Apparatus. As the reviewer correctly points out, each stimulus contained both luminance and chromaticity contrasts. However, both of these were kept constant over stimuli, while the only manipulated features were hue and motion direction, which we subsequently decoded from the neural data.

7) In the Discussion, please address reviewer 2's comment on bimodality. This reviewer highlights a possibility that direction columns have a spacing much smaller than that for axis of motion, a possibility raised by the early columnar models in macaque (Albright et al., 1984; Born and Bradley, 2005). A possibility supported by high resolution fMRI data that the "axis of motion" columns in human MT/V5 may include neurons of opposite direction selectivity (Zimmerman et al., 2011; Schneider et al., 2019). If the direction columns are significantly smaller than those of axis of motion, the source localization and measurements of bimodality may differ from each other in MT and other areas. In this context, the work of An et al., 2012 pointing to sources of motion signals in cortical areas outside MT should also be considered.

We thank the reviewer for bringing up this important possibility. We added a paragraph to the Discussion section exploring the consequences of columns with opposite motion direction selectivity being in close proximity in MT (subsection “Early ventral stream areas as sources of non-invasive information”, third paragraph). Briefly, we consider it unlikely that the bimodally tuned motion information measures in EEG/MEG was mainly driven by MT activity, but we point out that our data does not support a definitive answer to this question.

8) In Figure 2, please change blue and green colors for different cortical areas, to make the easier to distinguish and improve the visibility of the two lines in Figure 2A.

We exchanged the colors to enhance the differences between cortical areas, while maintaining our overall goals of a consistent color scheme and of a continuous color gradient. Additionally, we moved the least visible line into the foreground, making it somewhat more visible. However, as the two lines are very similar, full visibility is challenging. We therefore added a note to the figure legend.

9) Provide the missing details highlighted by reviewer 1.

We added the missing details, specifically we 1) explained the distinction between error bars over sessions in invasive macaque recordings and monkey EEG recordings, and error bars over participants in human MEG, 2) added information about the unrelated delayed saccade task (Materials and methods / Macaque microelectrode recordings / Stimuli and Apparatus), 3) made the requested changes to the color scheme / line visibility.

Complete reviews for your information:Reviewer #1:[…] I think there are some limitations of the study which should be discussed, and I have also identified some issues which need clarification.1) The decoding accuracy for single-unit and multi-unit activity is very low and this puzzles me. Let's take MT as an example: coding accuracy is well below 0.2 in all cases (chance level is at 0.125). We know that the majority of neurons in MT is tuned to motion direction and 100% coherent random dot stimuli are perhaps the stimuli that best drives these neurons. Thus, I expected a decoding accuracy for 1 out of 8 directions close to 1.0 in MT. In fact, Mendoza-Halliday et al., 2014 reported an accuracy > 0.5 even in PFC. The single unit tuning curves in Figure 7F also show high selectivity. What explains the low accuracy in the manuscript (e.g. Figure 2)? Is it because of the bandpass filtering of spike trains? I think in order to properly relate invasive and non-invasive recording it must be assured that both measurements are consistent with previous literature.

See above (Essential revisions, point 1).

2) Statistical significance of decoding results. Most of the decoding accuracies shown in Figure 2 are really low, sometimes ~0.1255 to 0.126 when the chance level is 0.125. Frankly, I don't understand how such small effects can be significant given the number of subjects and trials. It is crucial to confirm that some of the tiny effects are not caused by confounds such as any small imbalance in the data that could be detected by the decoder (unequal trial numbers, differences between individuals, etc.). At the very least I would suggest running the whole decoding analysis on data with shuffled stimulus class labels. This should not yield significant decoding apart from a few false positives. More detailed questions concerning the statistical procedures: Why only a 2-fold cross validation for SUA and MUA and not 10-fold as for EEG and MEG? Cluster permutation procedure (subsection “Multivariate classification”, last paragraph): I do not understand the rationale of multiplying with +1 or -1. This should be explained better and a reference should be given. Finally, is the use of t-tests appropriate here?

See above (Essential revisions, point 2).

3) One limitation of the study is that subjects view the random dot patterns just passively. This may reduce the stimulus representation in higher brain areas (see e.g. work of Romo and Pasternak with active and passive tasks). This should be discussed.

See above (Essential revisions, point 3).

4) Data availability. The authors have put the data points that are necessary to plot the figures in a repository, but this is of limited use for the community. I would strongly suggest making the full data available for further analysis. I hope this request is consistent with the spirit of eLife. The main advance of this manuscript is methodological, and I believe that providing this data as a resource to the community would be a further argument for publishing it in eLife.

We thank the reviewer for the request and we agree that open data is desirable. Of course, we are happy to provide raw data upon reasonable request. However, due to the effort involved in collecting data of this kind and other ongoing projects, we refrain from publishing the full raw data without restriction at this time

Reviewer #2:General:The overall goal of synthesizing work from humans and macaques, and across different functional tools (e.g. EEG, electrophysiology and MEG), and across spatial scales, is very important. In that sense alone, this study is valuable and should have impact. Also, the analysis is relatively sophisticated and quantitative. However, the biological interpretation of the underlying neural selectivity is a little naïve, which weakens support for the overall conclusions. Presumably, the authors could mitigate the latter concern by revising the text accordingly.Specific:Based on the wealth of prior research in both macaques and humans, it is well accepted that most/many neurons in area MT/V5 respond to stimuli in a direction-selective manner. However it is important to remember that area MT/V5 is a relatively small area, and not located on the cortical surface (thus arguably less amenable to accurate source analysis). More importantly, many other cortical areas (in areas V1, V2, V3, V3A, MSTd/l, etc.) also have many direction-selective neurons (or direction-activated) neurons, and most of those areas are much larger compared to MT/V5. Thus it seems likely that such areas contribute significantly to the 'motion' contrast tested here; i.e. it was not produced entirely (or perhaps even preferentially) in area MT/V5.

See above (Essential revisions, point 4).

The attribution of color-selective responses to 'V4' is less certain. For one thing, the oval indicating the 'V4' sampling site in macaque (Figure 1C) is overlaid on foveal V1/V2, not V4. The confusion is partly due to the presence of a very small gyrus (between the operculum and lunate gyrus) that connects the foveal representations of V1, V2, V3 and V4. A second concern is that color selective neurons and columns/patches/responses are not confined to (nor prominently present in) area V4; that is a persistent myth based on a few recordings made in the 1970s. Subsequent research has demonstrated many color responsive columns/patches/response in many other areas, (although not in MT). Thirdly, the color stimulus in 1A appears to be a mix of black dots and colored dots – i.e. a stimulus which varies in both luminance and color. [However I could not find a direct description of the color stimulus – so I am not certain about the stimulus]. Fourth, the color selectivity in humans (e.g. the L*C*h* space) differs from that in macaques, partly because the ratio of long- to medium-wavelength cones varies by a factor of two between these two species.Thus, I conclude that the measured responses to these different stimuli may well differ as reported in the recording sites, consistent with the authors interpretation. However, the simple attribution of color-vs.-motion selectivity to MT and V4 needs to be significantly qualified. Presumably if the authors had been able to directly map the cortical sites that respond selectively to color or motion everywhere in visual cortex, the source localization would have been more certain.

For the concerns regarding area V4, see above (Essential revisions, point 5). We thank the reviewer for bringing up the concerns regarding color selectivity differences between species. While stimuli are often designed based on human color spaces for macaque experiments, and while color vision is quite similar in both species, we are aware of the differences mentioned by the reviewer. For the purposes of this study, we do not consider a slightly distorted color space very problematic, however the potential confound of color hue and luminance is an issue we would like to address. We therefore decided to include a supplementary figure (Figure 4—figure supplement 1) providing evidence that equiluminant colors from a human L*C*h – space also appear close to equiluminant to macaque monkeys.

The source localization of MEG and EEG information is well known to be roughly inverted, differentially maximal from sulci vs. gyri in the two measurements. This issue is crucial in the determination of source localization in brain. The authors might explore and discuss this in more detail; here I instead got the impression that the EEG and MEG signals were similar.The extensive discussion of bi-modality (vs. lack thereof) in color vs. motion may change during the revision. However, it may be relevant that more recent attempts to map 'direction' columns (i.e. a dimension of 360 degrees) in cortex have yielded only axis-of-motion columns (a 180 degree dimension). Also, as briefly discussed by the authors, attempts to model color-selective cortical responses as psychophysically-color-opponent signals have not been entirely successful. These backgrounds may clarify the bi-modality discussion.

See above (Essential revisions, point 7).